# Effect of Process Orientation on the Mechanical Behavior and Piezoelectricity of Electroactive Paper

**DOI:** 10.3390/ma13010204

**Published:** 2020-01-03

**Authors:** Sean Yoon, Jung Woong Kim, Hyun Chan Kim, Jaehwan Kim

**Affiliations:** 1DPAMSTECH Co., Ltd., 75 Malgeunnae-gil, Uiwang-si, Gyeonggi-do 16071, Korea; sjyoon@dpamstech.co.kr; 2CRC for Nanocellulose Future Composites, Department of Mechanical Engineering, Inha University, Incheon 22212, Korea; jw6294@naver.com (J.W.K.); Kim_HyunChan@naver.com (H.C.K.)

**Keywords:** electroactive paper, piezoelectricity, cellulose, wet drawing, finite element method

## Abstract

This paper reports the effect of process orientation on the mechanical behavior and piezoelectricity of electroactive paper (EAPap) made from natural cotton pulp. EAPap is fabricated by a casting and wet drawing of cellulose film after dissolving cotton with LiCl and DMAc solvent. During the fabrication, permanent wrinkles, a possible factor for performance deterioration, were found in the films. Finite element method was introduced to identify the formation mechanism behind the wrinkles. The simulation results show that the wrinkles were caused by buckling and are inevitable under any conditions. The tensile and piezoelectric tests show that the orientation dependency of the stretched EAPap gives the anisotropic characteristics on both mechanical and piezoelectric properties. In this research, the anisotropic elastic moduli and Poisson’s ratios are reported. The piezoelectric charge constant of EAPap in the linear elastic is calculated. The piezoelectric charge constants of EAPap are associated with the alignment angle in the order of 45° > 0° > 90° due to the strong shear effect. The higher stretching ratio gives the higher piezoelectricity due to the alignment of the molecular chains and the microstructure in EAPap. The highest piezoelectric charge constant is found to be 12 pC/N at a stretching ratio of 1.6 and aligning angle of 45°.

## 1. Introduction

Cellulose, one of the most abundant and environmentally friendly materials on the Earth, has been used in various areas for food, textile, and papermaking. Meanwhile, many previous studies discovered that cellulose has piezoelectricity [1,2,3]. These studies prompted the prospect of developing cellulose-based industrial actuators, sensors, and speakers. In fact, some studies reported electroactive response and sound pressure levels from smart cellulose film actuators and speakers [4,5,6]. This smart cellulose film is called electroactive paper (EAPap). Other materials, such as ceramics and polymers, also show piezoelectric characteristics [7,8]. The piezoelectric ceramic, however, is seldom labelled a biomaterial because of its use of lead (Pb). In addition to the use of lead in their fabrication, piezoelectric ceramics are too brittle. Thus, it is legitimate to say that cellulose may have potential as an eco-friendly electroactive material [9]. 

In the previous studies, despite the importance of the material deformation in EAPap, the orientation dependency on both mechanical and piezoelectric behavior has not been intensively studied. Some papers reported the microstructural changes in cellulose by using SEM (scanning electron microscope) [4,5,6]. They are, however, focused on the morphology immediately after the fabrication and did not correlate the detailed mechanical stress–strain response and piezoelectricity to the orientation. Due to the nature of electroactive material, the material deformation is strongly related to its piezoelectricity. The stress–strain behavior, surface morphology, and the formation of wrinkles in EAPap can be related to the mechanism behind the piezoelectricity characteristics. We are not aware of any previous paper reporting detailed mechanical and piezoelectric behaviors of EAPap.

In this paper, we report the effect of process orientation on the mechanical behavior and piezoelectricity of EAPap. EAPap was fabricated by a casting and wet drawing of cellulose film made with natural cotton pulp and LiCl/DMAc solvent. The film was stretched to enhance the mechanical properties and piezoelectricity. A theoretical model to analyze the roll-to-roll process for EAPap was reported, consisting of casting, curing, and drying processes, except stretching [10]. During the stretching process, permanent wrinkles, a possible piezoelectricity deterioration factor, were found in the EAPap. Since the piezoelectricity of EAPap was greatly affected by the alignment and mechanical strain of the molecular chains in the film, we introduced a finite element method to find the mechanism behind the wrinkles and possibly prevent the formation of wrinkles during fabrication. Tensile tests were then performed to find the stress–strain relation of EAPap. The anisotropic elastic moduli of stretched EAPap were calculated. A set of the piezoelectricity measurements as a function of stretching ratio and alignment angle were made to find the maximum piezoelectric charge constant because the alignment in the film and the corresponding mechanical behavior may be correlated.

## 2. Materials and Methods 

Figure 1 summarizes the fabrication process of EAPap by regenerating natural cotton pulp using a house-built system. The natural cotton pulp was provided by Georgia-Pacific Chemical (Atlanta, GA, USA). The raw cotton pulp was dried at 120 °C for 30 min in a heating oven and mixed with LiCl (Sigma-Aldrich, St. Louis, MO, USA). N, N-dimethylacetamide or DMAc (Sigma-Aldrich) was added to dissolve the mixture solution. The cellulose solution then was poured into a solution tank as shown in Figure 1. In the casting process, the film was subjected to a sprinkled mixture of water and isopropyl alcohol. The film then passed through multiple rollers in the drying system. Near the end of the drying process, the film was stretched to have stretching ratios from 1.5 to 1.8. Aluminum electrodes were deposited on both sides of the film using a sputtering machine. After the fabrication, the films were cut to have an area of 40 mm × 12 mm, with three orientation directions with respect to the stretching direction, 0°, 45°, and 90°. For simplicity, we denote these orientations as θ0, θ45, and θ90, respectively. The film thickness measured was approximately 20 μm. The film samples then were taken for tensile testing and piezoelectric testing. A tensile tester (DTT-701C, Daekyung Tech, Incheon, Korea) with a linear scaler (GB-BA/SR128–015, Sony, Tokyo, Japan,) and a picoammeter (Model6485, Keithley, Solon, OH, USA) were used. Figure 2 shows the schematic of piezoelectric charge constant measurement. Note that it is important to remove all space charges which accumulated during the fabrication beforehand by grounding. The amount of induced charge was calculated by integrating the measured current. The in-plane piezoelectric charge constant then is calculated as
(1)$$d_{31}=\frac{Q\;\cdot \;A_{c}}{F\cdot A_{e}}$$
where Q, A_c_, F, and A_e_ denote the induced charges, the cross-sectional area, the applied load, and the area of the electrode, respectively.

## 3. Results and Discussion

This section is divided by subheadings. It provides a concise and precise description of the experimental results and their interpretation as well as the experimental conclusions that can be drawn.

First, it is noteworthy that permanent wrinkles showed up in the films during the fabrication, as shown in Figure 3. The formation of wrinkles is important because the piezoelectricity may be deteriorated by the modified molecular chain orientations. For reliable piezoelectricity, EAPap needs to maintain a consistent surface, possibly with no or minimized wrinkles. In order to find how the wrinkles were formed or how to remove them, we simulated the formation of wrinkles with a finite element analysis code, ANSYS (ANSYS, Canonsburg, PA, USA).

Figure 4 shows the simulated formation of wrinkles during the stretching process. The parameters used are summarized in Table 1. Displacement boundary conditions are specified at two rollers associated with the velocity difference between two rollers. The alignment mismatch between the rollers induces the shear stress adding to the tension in the longitudinal direction. The wrinkles are inclined with an angle of 7.5° depending on the mismatch. The wavelength is found to be 26.86 mm. The out-of-plane displacement increases gradually from the edge while oscillating alternately and reaches its maximum near the center. The wrinkle at the bottom shifts its peak to the right while the wrinkle in the top shifts its peak to the left.

Figure 5 shows the simulated formation of wrinkles in EAPap without the alignment mismatch. The simulation parameters and properties are equal to those in Figure 4 except the alignment mismatch. The wrinkles did not have inclinations as shown in Figure 4. The wavelength was found to be 33.3 mm. The removal of the shear strain apparently increases the wavelength.

The results shown in Figure 4 and Figure 5 suggest that whether the roller is misaligned or not, the shear strain always occurs, and the film is subject to buckling. Thus, we conclude that it is virtually impossible to remove all the wrinkles. Previous studies [11,12,13] obtained similar conclusions for other polymer films. One possible solution in restraining the formation of wrinkles is, instead of the in-line fabrication we employed, to use the additional constraining on the edges along the lateral direction. In that case, the strains that cause buckling will be restrained and remove the wrinkles. However, installing the additional constraining to the in-line fabrication system is very difficult because the film is constantly moving.

Figure 6 shows the stress–strain curves of stretched EAPap with the orientation of θ0, θ45, and θ90. At the very low strains (<0.005), the stress increased very slowly. The stress–strain relationship in this region resembles reversible nonlinear viscoelastic behavior. The stress afterward increases rapidly in a linear fashion in all three, θ0, θ45, and θ90 cases. The maximum tensile strength is shown in the θ0 case and the minimum value in the θ90 case. This orientation dependency of stretched EAPap may not be surprising given the re-alignment of the cellulose molecule chains which occurs during the stretching process. Table 2 shows the mechanical properties of stretched EAPap. In Table 2, 1 and 2 denote the test directions corresponding to the stretching directions of 0° and 90° in Figure 1. The tensile tests were equipped with several microscopes, gauges, and load cells to measure elastic moduli for the planar orthogonal case. Note that several other studies reported isotropic properties for general-purpose cellulose films [14,15], which imply that the orientation dependency was created in the stretched EAPap.

Figure 7 shows the measured induced charge as a function of load. From the graphs, it is clear that the electric charges in elastic and plastic regions have different charge/load slopes. The electrical charges in a linear elastic region did not dramatically increase as the load increased. The induced charges in the plastic region, however, rapidly increased as the load increased. The film may have the strong plastic region dependency on accumulating electric charges. However, we use the induced charges in the linear elastic region to calculate piezoelectric charge constants because the typical piezoelectric charge constant is defined in the linear region.

Table 3 shows the piezoelectric charge constants of EAPap fabricated with various stretching ratios and aligning angles. Piezoelectric charge constants, d_31_, for θ0, θ45, and θ90 are 7.3, 12.0, and 1.5 pC/N in the order of θ45 > θ90 > θ0, which implies the strong shear effect. This strong shear effect was previously studied [16] using different types of films. Piezoelectric charge constant for the θ0 case as a function of increased stretching ratio (from 1.5 to 1.8) increased from 5.9 to 10.1 pC/N. The mechanism behind the piezoelectricity of stretched EAPap can be delineated as the relation between the dipole creation and the crystalline structure [4,17] as follows: EAPap has both ordered and disordered domains. The ordered domain is mostly crystalline while the disordered domain is more amorphous. In the view of chemical bonding, EAPap was produced with enough hydrogen bonds coming from water and chlorine ions. The stretching process induced the alignment of the monocrystalline structures and the fibrils. Cations and anions then formed the dipoles between the molecule chains, which induced more piezoelectric charges. In this case, the alignment of the molecular chains and the microstructure of the film are able to enhance the piezoelectricity in cellulose films.

## 4. Conclusions

In this study, electroactive paper (EAPap) was fabricated using a wet drawing method. Cellulose fibers from natural raw cotton pulp were dissolved by using LiCl and DMAc solvent and then generated cellulose films. The wet drawing method was able to increase the alignment of cellulose film. The wrinkles, a possible factor for piezoelectricity deterioration, were found in the films during the fabrication. Finite element analysis showed that the wrinkles stem from buckling, which is permanent and inevitable. The stress–strain curves of stretched EAPap showed the orientation dependency. The anisotropy of stretched EAPap was observed as an orthogonal material and calculated the anisotropic properties. The piezoelectricity measurement showed that the piezoelectric charge constants were as a function of the alignment angle in the order of 45° > 0° > 90°. The higher stretching ratio produced the higher piezoelectricity. The highest piezoelectric charge constant was found to be 12.0 pC/N at the stretching ratio of 1.6 and the aligning angle of 45°. This implies that the stretching process improved the piezoelectricity and the strong shear piezoelectricity effect. We concluded that the outstanding piezoelectric performance of EAPap can be used as a piezoelectric material for various industrial applications.

## Figures and Tables

**Figure 1 materials-13-00204-f001:**
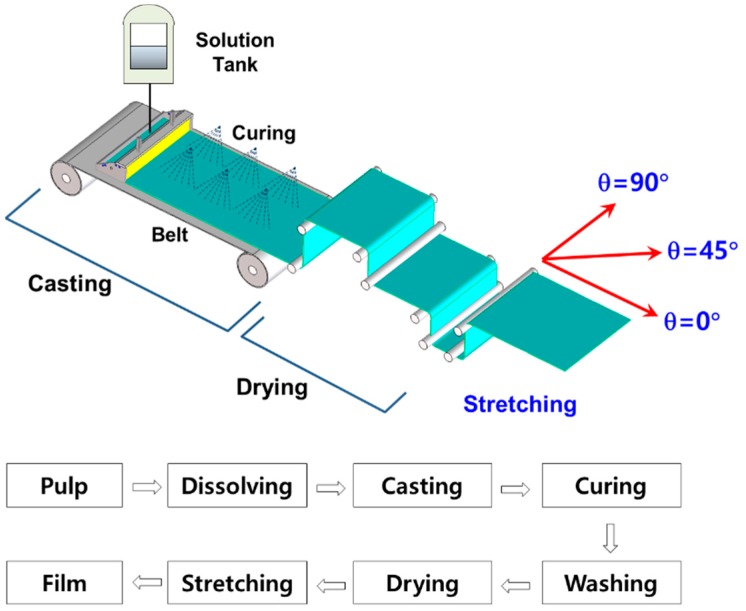
Fabrication process of electroactive paper (EAPap).

**Figure 2 materials-13-00204-f002:**
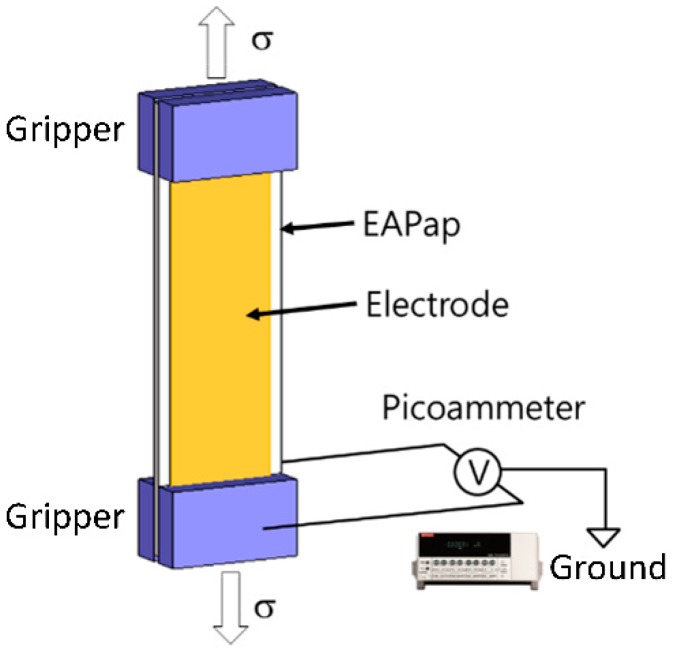
Measurement of piezoelectric charge constant.

**Figure 3 materials-13-00204-f003:**
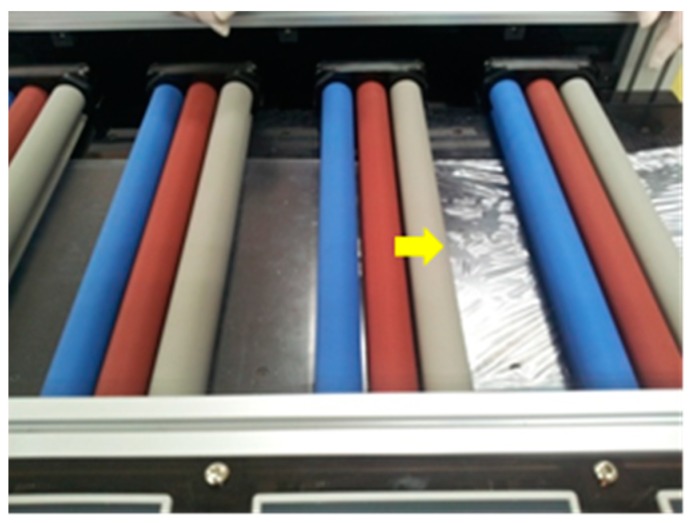
Formation of wrinkles in EAPap during the drying process.

**Figure 4 materials-13-00204-f004:**
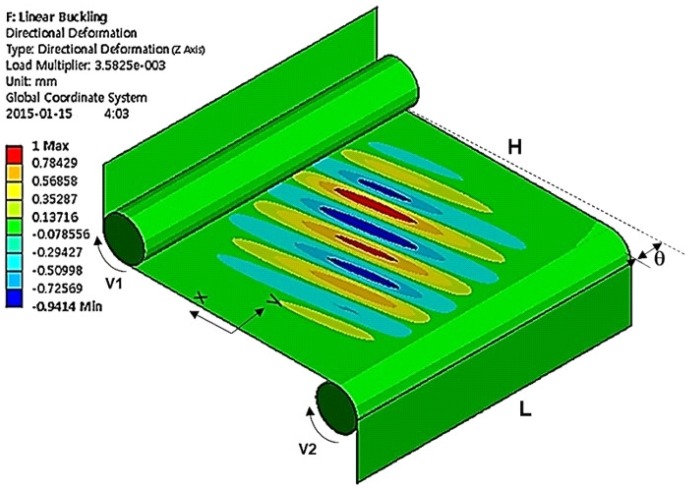
Simulated formation of wrinkles with an alignment mismatch.

**Figure 5 materials-13-00204-f005:**
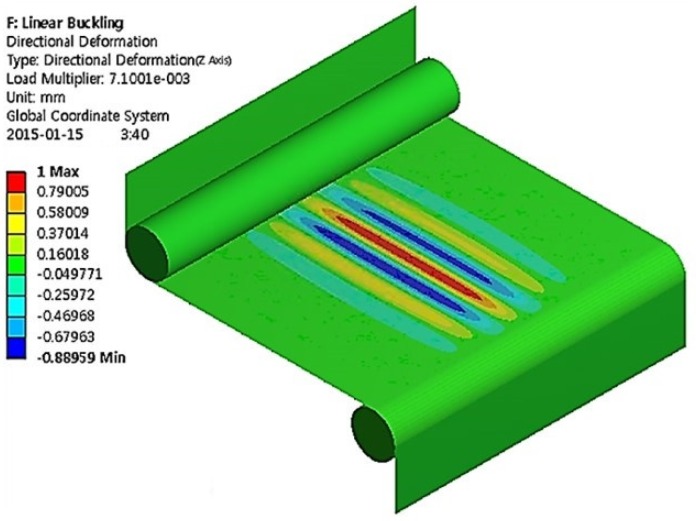
Simulated formation of wrinkles without an alignment mismatch.

**Figure 6 materials-13-00204-f006:**
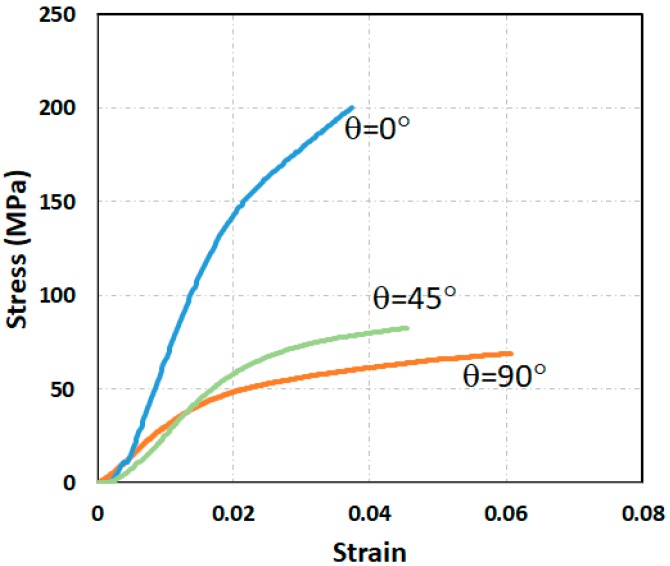
Stress–strain curves of stretched EAPap with the orientation of 0°, 45°, and 90°.

**Figure 7 materials-13-00204-f007:**
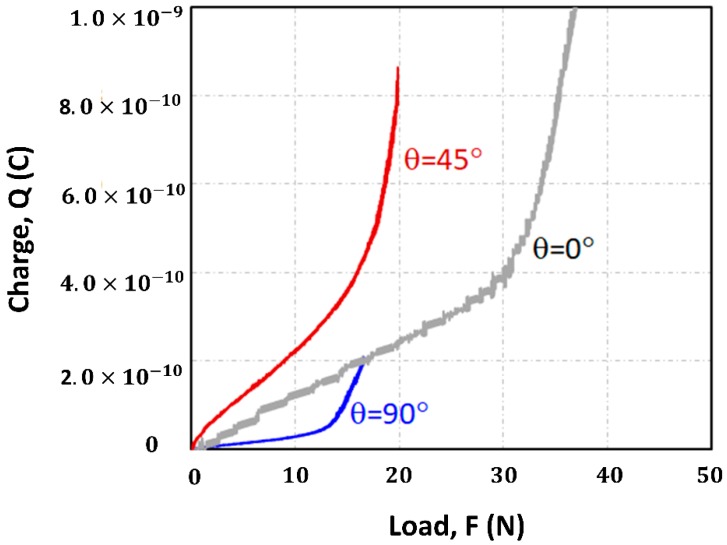
Induced charge as a function of strain of load of EAPap.

**Table 1 materials-13-00204-t001:** Simulation parameters used in this study for the stretching.

Parameter	Unit	Value
Film thickness	mm	0.02
Film width	mm	200
Film span length	mm	330
Roller Speed V1	rpm	1.50
Roller Speed V2	rpm	1.76
Mismatch θ	°	1.6

**Table 2 materials-13-00204-t002:** Mechanical properties of stretched EAPap.

Modulus	Unit	Value
*E* _1_	MPa	10,265
*E* _2_	MPa	3983
*ν* _12_	–	0.389
*ν* _21_	–	0.151

**Table 3 materials-13-00204-t003:** Piezoelectricity charge constants of various EAPap.

Stretching Ratio	Alignment Angle (θ)	Piezoelectricity (pC/N)
1.5	0	5.9
1.6	0	7.3
1.6	45	12.0
1.6	90	1.5
1.8	0	10.1

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
