# Peer review of "Effect of Process Orientation on the Mechanical Behavior and Piezoelectricity of Electroactive Paper"

_materials, 2020, doi:10.3390/ma13010204_

Round 1

Reviewer 1 Report

This article could be accepted for publication if the authors resolve the following weaknesses:

Line 37, p. 1: please replace the word “echo” with “eco”!

Line 72, p. 2: I think this is a flat sample with dimensions of 40x12 mm; it is not a “cross section”.

Line 94, p 3:  please correct the English: “the formation is important” or “the formation of wrinkles is important” ….   

Figs. 4 and 5: the applied load value is not mentioned, the film cannot be tensioned only due to difference between roller's speed (v1 and v2). How did you apply the load? It is not clear, since they are several ways to do it.

Author Response

This article could be accepted for publication if the authors resolve the following weaknesses:

Answer=> Thank you.  

Line 37, p. 1: please replace the word “echo” with “eco”!

Answer=> Thank you for the comment. It was corrected.

Line 72, p. 2: I think this is a flat sample with dimensions of 40x12 mm; it is not a “cross section”.

Answer=> Thank you for the comment. Yes it was area, not cross section. It was corrected.

Line 94, p 3:  please correct the English: “the formation is important” or “the formation of wrinkles is important” ….  

Answer=> Thank you for the comment. The sentence was corrected as suggested.

Figs. 4 and 5: the applied load value is not mentioned, the film cannot be tensioned only due to difference between roller's speed (v1 and v2). How did you apply the load? It is not clear, since they are several ways to do it.

Answer=> Thank you for the comment. The boundary conditions at two rollers are not force boundary conditions. They are displacement boundary conditions associated with the velocity difference between two rollers. In Page 4, the following sentence was included: “Displacement boundary conditions are specified at two rollers associated with the velocity difference between two rollers.”

Reviewer 2 Report

In the manuscript, the authors describe the new orientation method and mechanical properties of a paper made of cellulose for piezoelectric papers. These results will be helpful and informative for the researchers in the field of materials chemistry.

Whereas the reviewer thinks that the authors’ study in this manuscript is interesting, suggestive, and well-organized, some descriptions are not enough. The authors’ manuscript is not suitable for publication in “Materials in the present form.

From these considerations, the reviewer recommends accepting for publication in " Materials," if the following issues are resolved.

How are the orientation and anisotropic properties of the authors’ material except for its mechanical properties? On the piezoelectric properties, it is important to evaluate the frequency dependence of materials on the applied voltage. What is the difference between the authors’ new method and the general method? The research data should be described and discussed in the manuscript.

Author Response

In the manuscript, the authors describe the new orientation method and mechanical properties of a paper made of cellulose for piezoelectric papers. These results will be helpful and informative for the researchers in the field of materials chemistry.

Whereas the reviewer thinks that the authors’ study in this manuscript is interesting, suggestive, and well-organized, some descriptions are not enough. The authors’ manuscript is not suitable for publication in “Materials” in the present form.

From these considerations, the reviewer recommends accepting for publication in "Materials," if the following issues are resolved.

Answer=> Thank you for the comments.

How are the orientation and anisotropic properties of the authors’ material except for its mechanical properties? On the piezoelectric properties, it is important to evaluate the frequency dependence of materials on the applied voltage. What is the difference between the authors’ new method and the general method? The research data should be described and discussed in the manuscript.

Answer=> Thank you for the comment. Of course this material has anisotropic behaviors in mechanical, dielectric and piezoelectric properties. There are several methods to measure piezoelectric properties: static and quasi-static measurements, resonator measurement and plane-wave velocity measurement [1]. This paper represented a quasi-static measurement method for measuring piezoelectricity of EAPap. Under tensile load, transverse electric charge was measured, which results in d31. It is still controversial which method is appropriate for measuring piezoelectric properties: the quasi-static method or resonator method. Piezoceramics are well characterized by resonant method and they are well applied for ultrasonic applications. Since the application of EAPap is low frequency, we believe that quasi-static measurement may be appropriate.

[1] Standards Committee of the IEEE Ultrasonics, Ferroelectrics, and Frequency Control Society, “IEEE Standard on Piezoelectricity,” ANSI/IEEE Std 176-1987, The Institute of Electrical and Electronics Engineers, Inc. 1988.

Round 2

Reviewer 2 Report

Whereas the reviewer understands the authors’ thoughts on the piezoelectric properties, some descriptions are not enough.

Why didn’t the authors answer the reviewer’s questions below?

(1) How are the orientation and anisotropic properties of the authors’ material except for its mechanical properties?

(2) What is the difference between the authors’ new method and the general method? The research data should be described and discussed in the manuscript.

What is new?

Author Response

How are the orientation and anisotropic properties of the authors’ material except for its mechanical properties? On the piezoelectric properties, it is important to evaluate the frequency dependence of materials on the applied voltage. What is the difference between the authors’ new method and the general method? The research data should be described and discussed in the manuscript.

Answer=> Thank you for the comment. Of course this material has anisotropic behaviors in mechanical, dielectric and piezoelectric properties. There are several methods to measure piezoelectric properties: static and quasi-static measurements, resonator measurement and plane-wave velocity measurement [1]. This paper represented a quasi-static measurement method for measuring piezoelectricity of EAPap. Under tensile load, transverse electric charge was measured, which results in d31. It is still controversial which method is appropriate for measuring piezoelectric properties: the quasi-static method or resonator method. Piezoceramics are well characterized by resonant method and they are well applied for ultrasonic applications. Since the application of EAPap is low frequency, we believe that quasi-static measurement may be appropriate.

[1] Standards Committee of the IEEE Ultrasonics, Ferroelectrics, and Frequency Control Society, “IEEE Standard on Piezoelectricity,” ANSI/IEEE Std 176-1987, The Institute of Electrical and Electronics Engineers, Inc. 1988.